# Privacy Odometers and Filters: Pay-as-you-Go Composition

Ryan Rogers[*]     Aaron Roth[†]     Jonathan Ullman[‡]     Salil Vadhan[§]

## Abstract

In this paper we initiate the study of adaptive composition in differential privacy when the length of the composition, and the privacy parameters themselves can be chosen adaptively, as a function of the outcome of previously run analyses. This case is much more delicate than the setting covered by existing composition theorems, in which the algorithms themselves can be chosen adaptively, but the privacy parameters must be fixed up front. Indeed, it isn't even clear how to define differential privacy in the adaptive parameter setting. We proceed by defining two objects which cover the two main use cases of composition theorems. A *privacy filter* is a stopping time rule that allows an analyst to halt a computation before his pre-specified privacy budget is exceeded. A *privacy odometer* allows the analyst to track realized privacy loss as he goes, without needing to pre-specify a privacy budget. We show that unlike the case in which privacy parameters are fixed, in the adaptive parameter setting, these two use cases are distinct. We show that there exist privacy filters with bounds comparable (up to constants) with existing privacy composition theorems. We also give a privacy odometer that nearly matches non-adaptive private composition theorems, but is sometimes worse by a small asymptotic factor. Moreover, we show that this is inherent, and that *any* valid privacy odometer in the adaptive parameter setting must lose this factor, which shows a formal separation between the filter and odometer use-cases.

## 1   Introduction

*Differential privacy* [DMNS06] is a stability condition on a randomized algorithm, designed to guarantee individual-level privacy during data analysis. Informally, an algorithm is differentially private if any pair of close inputs map to similar probability distributions over outputs, where similarity is measured by two parameters $\varepsilon$ and $\delta$. Informally, $\varepsilon$ measures the *amount of privacy* and $\delta$ measures the *failure probability* that the privacy loss is much worse than $\varepsilon$. A signature property of differential privacy is that it is preserved under *composition*—combining many differentially private subroutines into a single algorithm preserves differential privacy and the privacy parameters degrade gracefully. Composability is essential for both privacy and for algorithm design. Since differential privacy is composable, we can design a sophisticated algorithm and prove it is private without having to rea-

---

[*]Department of Applied Mathematics and Computational Science, University of Pennsylvania. ryrogers@sas.upenn.edu.

[†]Department of Computer and Information Sciences, University of Pennsylvania. aaroth@cis.upenn.edu. Supported in part by an NSF CAREER award, NSF grant CNS-1513694, and a grant from the Sloan Foundation.

[‡]College of Computer and Information Science, Northeastern University. jullman@ccs.neu.edu

[§]Center for Research on Computation & Society and John A. Paulson School of Engineering & Applied Sciences, Harvard University. salil@seas.harvard.edu. Work done while visiting the Department of Applied Mathematics and the Shing-Tung Yau Center at National Chiao-Tung University in Taiwan. Also supported by NSF grant CNS-1237235, a grant from the Sloan Foundation, and a Simons Investigator Award.

son directly about its output distribution. Instead, we can rely on the differential privacy of the basic building blocks and derive a privacy bound on the whole algorithm using the composition rules.

The composition theorem for differential privacy is very strong, and holds even if the choice of which differentially private subroutine to run is adaptive—that is, the choice of the next algorithm may depend on the output of previous algorithms. This property is essential in algorithm design, but also more generally in modeling unstructured sequences of data analyses that might be run by a human data analyst, or even by many data analysts on the same data set, while only loosely coordinating with one another. Even setting aside privacy, it can be very challenging to analyze the statistical properties of general adaptive procedures for analyzing a dataset, and the fact that adaptively chosen differentially private algorithms compose has recently been used to give strong guarantees of statistical validity for adaptive data analysis [DFH+15, BNS+16].

However, all the known composition theorems for differential privacy [DMNS06, DKM+06, DRV10, KOV15, MV16] have an important and generally overlooked caveat. Although the choice of the next subroutine in the composition may be adaptive, the number of subroutines called and choice of the privacy parameters $\varepsilon$ and $\delta$ for each subroutine must be fixed in advance. Indeed, it is not even clear how to define differential privacy if the privacy parameters are not fixed in advance. This is generally acceptable when designing a single algorithm (that has a worst-case analysis), since worst-case eventualities need to be anticipated and budgeted for in order to prove a theorem. However, it is *not* acceptable when modeling the unstructured adaptivity of a data analyst, who may not know ahead of time (before seeing the results of intermediate analyses) what he wants to do with the data. When controlling privacy loss across multiple data analysts, the problem is even worse.

As a simple stylized example, suppose that $\mathcal{A}$ is some algorithm (possibly modeling a human data analyst) for selecting *statistical queries*[5] as a function of the answers to previously selected queries. It is known that for any one statistical query $q$ and any data set $\mathbf{x}$, releasing the perturbed answer $\hat{a} = q(\mathbf{x}) + Z$ where $Z \sim \mathrm{Lap}(1/\varepsilon)$ is a Laplace random variable, ensures $(\varepsilon, 0)$-differential privacy. Composition theorems allow us to reason about the composition of $k$ such operations, where the queries can be chosen adaptively by $\mathcal{A}$, as in the following simple program.

**Example1**$(\mathbf{x})$:
    For $i = 1$ to $k$: Let $q_i = \mathcal{A}(\hat{a}_1, \dots, \hat{a}_{i-1})$ and let $\hat{a}_i = q_i(\mathbf{x}) + \mathrm{Lap}(1/\varepsilon)$.
    Output $(\hat{a}_1, \dots, \hat{a}_k)$.

The "basic" composition theorem [DMNS06] asserts that Example1 is $(\varepsilon k, 0)$-differentially private. The "advanced" composition theorem [DRV10] gives a more sophisticated bound and asserts that (provided that $\varepsilon$ is sufficiently small), the algorithm satisfies $(\varepsilon\sqrt{8k\ln(1/\delta)}, \delta)$-differential privacy for any $\delta > 0$. There is even an "optimal" composition theorem [KOV15] too complicated to describe here. These analyses crucially assume that both the number of iterations $k$ and the parameter $\varepsilon$ are fixed up front, even though it allows for the queries $q_i$ to be adaptively chosen.[6]

Now consider a similar example where the number of iterations is not fixed up front, but actually depends on the answers to previous queries. This is a special case of a more general setting where the privacy parameter $\varepsilon_i$ in every round may be chosen adaptively—halting in our example is equivalent to setting $\varepsilon_i = 0$ in all future rounds.

**Example2**$(\mathbf{x}, \tau)$:
    Let $i \leftarrow 1$, $\hat{a}_1 \leftarrow q_1(\mathbf{x}) + \mathrm{Lap}(1/\varepsilon)$.
    While $\hat{a}_i \leq \tau$: Let $i \leftarrow i + 1$, $q_i = \mathcal{A}(\hat{a}_1, \dots, \hat{a}_{i-1})$, and let $\hat{a}_i = q_i(\mathbf{x}) + \mathrm{Lap}(1/\varepsilon)$.
    Output $(\hat{a}_1, \dots, \hat{a}_i)$.

Example2 cannot be said to be differentially private *ex ante* for any non-trivial fixed values of $\varepsilon$ and $\delta$, because the computation might run for an arbitrarily long time and privacy may degrade indefinitely. What can we say about privacy after we run the algorithm? If the algorithm/data-analyst happens to stop after $k$ rounds, can we apply the composition theorem *ex post* to conclude that it is $(\varepsilon k, 0)$- and

$(\varepsilon\sqrt{8k\log(1/\delta)}, \delta)$-differentially private, as we could if the algorithm were constrained to always run for at most $k$ rounds?

In this paper, we study the composition properties of differential privacy when *everything*—the choice of algorithms, the number of rounds, and the privacy parameters in each round—may be adaptively chosen. We show that this setting is much more delicate than the settings covered by previously known composition theorems, but that these sorts of *ex post* privacy bounds do hold with only a small (but in some cases unavoidable) loss over the standard setting. We note that the conceptual discussion of differential privacy focuses a lot on the idea of arbitrary composition and our results give more support for this conceptual interpretation.

## 1.1 Our Results

We give a formal framework for reasoning about the adaptive composition of differentially private algorithms when the privacy parameters themselves can be chosen adaptively. When the parameters are chosen non-adaptively, a *composition theorem* gives a high probability bound on the worst case *privacy loss* that results from the output of an algorithm. In the adaptive parameter setting, it no longer makes sense to have fixed bounds on the privacy loss. Instead, we propose two kinds of primitives capturing two natural use cases for composition theorems:

1. A *privacy odometer* takes as input a global failure parameter $\delta_g$. After every round $i$ in the composition of differentially private algorithms, the odometer outputs a number $\tau_i$ that may depend on the *realized* privacy parameters $\varepsilon_i, \delta_i$ in the previous rounds. The privacy odometer guarantees that with probability $1 - \delta_g$, for every round $i$, $\tau_i$ is an upper bound on the privacy loss in round $i$.

2. A *privacy filter* is a way to cut off access to the dataset when the privacy loss is too large. It takes as input a global privacy "budget" $(\varepsilon_g, \delta_g)$. After every round, it either outputs CONT ("continue") or HALT depending on the privacy parameters from the previous rounds. The privacy filter guarantees that with probability $1 - \delta_g$, it will output HALT before the privacy loss exceeds $\varepsilon_g$. When used, it guarantees that the resulting interaction is $(\varepsilon_g, \delta_g)$-DP.

A tempting heuristic is to take the *realized* privacy parameters $\varepsilon_1, \delta_1, \ldots, \varepsilon_i, \delta_i$ and apply one of the existing composition theorems to those parameters, using that value as a privacy odometer or implementing a privacy filter by halting when getting a value that exceeds the global budget. However this heuristic *does not* necessarily give valid bounds.

We first prove that the heuristic *does* work for the basic composition theorem [DMNS06] in which the parameters $\varepsilon_i$ and $\delta_i$ add up. We prove that summing the realized privacy parameters yields both a valid privacy odometer and filter. The idea of a privacy filter was also considered in [ES15], who show that basic composition works in the privacy filter application.

We then show that the heuristic breaks for the advanced composition theorem [DRV10]. However, we give a valid privacy filter that gives the same asymptotic bound as the advanced composition theorem, albeit with worse constants. On the other hand, we show that, in some parameter regimes, the asymptotic bounds given by our privacy filter *cannot* be achieved by a privacy odometer. This result gives a formal separation between the two models when the parameters may be chosen adaptively, which does not exist when the privacy parameters are fixed. Finally, we give a valid privacy odometer with a bound that is only slightly worse asymptotically than the bound that the advanced composition theorem would give if it were used (improperly) as a heuristic. Our bound is worse by a factor that is never larger than $\sqrt{\log\log(n)}$ (here, $n$ is the size of the dataset) and for some parameter regimes is only a constant.

## 2 Privacy Preliminaries

Differential privacy is defined based on the following notion of similarity between two distributions.

**Definition 2.1** (Indistinguishable)**.** Two random variables $X$ and $Y$ taking values from domain $\mathcal{D}$ are $(\varepsilon, \delta)$-indistinguishable, denoted as $X \approx_{\varepsilon,\delta} Y$, if $\forall S \subseteq \mathcal{D}$, $\mathbb{P}[X \in S] \leq e^\varepsilon \mathbb{P}[Y \in S] + \delta$ and $\mathbb{P}[Y \in S] \leq e^\varepsilon \mathbb{P}[X \in S] + \delta$.

There is a slight variant of indistinguishability, called *point-wise indistinguishability*, which is nearly equivalent, but will be the more convenient notion for the generalizations we give in this paper.

**Definition 2.2** (Point-wise Indistinguishable). Two random variables $X$ and $Y$ taking values from $\mathcal{D}$ are $(\varepsilon, \delta)$-point-wise indistinguishable if with probability at least $1 - \delta$ over either $a \sim X$ or $a \sim Y$, we have $\left| \log \left( \frac{\mathbb{P}[X=a]}{\mathbb{P}[Y=a]} \right) \right| \leq \varepsilon$.

**Lemma 2.3** ([KS14]). *Let $X$ and $Y$ be two random variables taking values from $\mathcal{D}$. If $X$ and $Y$ are $(\varepsilon, \delta)$-point-wise indistinguishable, then $X \approx_{\varepsilon, \delta} Y$. Also, if $X \approx_{\varepsilon, \delta} Y$ then $X$ and $Y$ are $\left(2\varepsilon, \frac{2\delta}{e^{\varepsilon}\varepsilon}\right)$-point-wise indistinguishable.*

We say two databases $\mathbf{x}, \mathbf{x}' \in \mathcal{X}^n$ are *neighboring* if they differ in at most one entry, i.e. if there exists an index $i \in [n]$ such that $\mathbf{x}_{-i} = \mathbf{x}'_{-i}$. We can now state differential privacy in terms of indistinguishability.

**Definition 2.4** (Differential Privacy [DMNS06]). A randomized algorithm $\mathcal{M} : \mathcal{X}^n \to \mathcal{Y}$ with arbitrary output range $\mathcal{Y}$ is $(\varepsilon, \delta)$-differentially private (DP) if for every pair of neighboring databases $\mathbf{x}, \mathbf{x}'$: $\mathcal{M}(\mathbf{x}) \approx_{\varepsilon, \delta} \mathcal{M}(\mathbf{x}')$.

We then define the privacy loss $\text{Loss}_{\mathcal{M}}(a; \mathbf{x}, \mathbf{x}')$ for outcome $a \in \mathcal{Y}$ and neighboring datasets $\mathbf{x}, \mathbf{x}' \in \mathcal{X}^n$ as $\text{Loss}_{\mathcal{M}}(a; \mathbf{x}, \mathbf{x}') = \log \left( \frac{\mathbb{P}[\mathcal{M}(\mathbf{x})=a]}{\mathbb{P}[\mathcal{M}(\mathbf{x}')=a]} \right)$. We note that if we can bound $\text{Loss}_{\mathcal{M}}(a; \mathbf{x}, \mathbf{x}')$ for any neighboring datasets $\mathbf{x}, \mathbf{x}'$ with high probability over $a \sim \mathcal{M}(\mathbf{x})$, then Theorem 2.3 tells us that $\mathcal{M}$ is differentially private. Moreover, Theorem 2.3 also implies that this approach is without loss of generality (up to a small difference in the parameters). Thus, our composition theorems will focus on bounding the privacy loss with high probability.

A useful property of differential privacy is that it is preserved under post-processing without degrading the parameters:

**Theorem 2.5** (Post-Processing [DMNS06]). *Let $\mathcal{M} : \mathcal{X}^n \to \mathcal{Y}$ be $(\varepsilon, \delta)$-DP and $f : \mathcal{Y} \to \mathcal{Y}'$ be any randomized algorithm. Then $f \circ \mathcal{M} : \mathcal{X}^n \to \mathcal{Y}'$ is $(\varepsilon, \delta)$-DP.*

We next recall a useful characterization from [KOV15]: any DP algorithm can be written as the post-processing of a simple, canonical algorithm which is a generalization of *randomized response*.

**Definition 2.6.** For any $\varepsilon, \delta \geq 0$, we define the *randomized response* algorithm $\text{RR}_{\varepsilon, \delta} : \{0, 1\} \to \{0, \top, \bot, 1\}$ as the following (Note that if $\delta = 0$, we will simply write the algorithm $\text{RR}_{\varepsilon, \delta}$ as $\text{RR}_{\varepsilon}$.)

$$\begin{aligned}
\mathbb{P}\left[\text{RR}_{\varepsilon, \delta}(0) = 0\right] &= \delta & \mathbb{P}\left[\text{RR}_{\varepsilon, \delta}(1) = 0\right] &= 0 \\
\mathbb{P}\left[\text{RR}_{\varepsilon, \delta}(0) = \top\right] &= (1 - \delta)\frac{e^{\varepsilon}}{1+e^{\varepsilon}} & \mathbb{P}\left[\text{RR}_{\varepsilon, \delta}(1) = \top\right] &= (1 - \delta)\frac{1}{1+e^{\varepsilon}} \\
\mathbb{P}\left[\text{RR}_{\varepsilon, \delta}(0) = \bot\right] &= (1 - \delta)\frac{1}{1+e^{\varepsilon}} & \mathbb{P}\left[\text{RR}_{\varepsilon, \delta}(1) = \bot\right] &= (1 - \delta)\frac{e^{\varepsilon}}{1+e^{\varepsilon}} \\
\mathbb{P}\left[\text{RR}_{\varepsilon, \delta}(0) = 1\right] &= 0 & \mathbb{P}\left[\text{RR}_{\varepsilon, \delta}(1) = 1\right] &= \delta
\end{aligned}$$

Kairouz, Oh, and Viswanath [KOV15] show that any $(\varepsilon, \delta)$–DP algorithm can be viewed as a post-processing of the output of $\text{RR}_{\varepsilon, \delta}$ for an appropriately chosen input.

**Theorem 2.7** ([KOV15], see also [MV16]). *For every $(\varepsilon, \delta)$-DP algorithm $\mathcal{M}$ and for all neighboring databases $\mathbf{x}^0$ and $\mathbf{x}^1$, there exists a randomized algorithm $T$ where $T(\text{RR}_{\varepsilon, \delta}(b))$ is identically distributed to $\mathcal{M}(\mathbf{x}^b)$ for $b \in \{0, 1\}$.*

This theorem will be useful in our analyses, because it allows us to without loss of generality analyze compositions of these simple algorithms $\text{RR}_{\varepsilon, \delta}$ with varying privacy parameters.

We now define the adaptive composition of differentially private algorithms in the setting introduced by [DRV10] and then extended to *heterogenous* privacy parameters in [MV16], in which all of the privacy parameters are fixed prior to the start of the computation. The following "composition game" is an abstract model of composition in which an adversary can adaptively select between neighboring datasets at each round, as well as a differentially private algorithm to run at each round – both choices can be a function of the realized outcomes of all previous rounds. However, crucially, the adversary must select at each round an algorithm that satisfies the privacy parameters which have been fixed ahead of time – the choice of parameters cannot itself be a function of the realized outcomes of previous rounds. We define this model of interaction formally in Algorithm 1 where the output is the *view* of the adversary $\mathcal{A}$ which includes any random coins she uses $R_{\mathcal{A}}$ and the outcomes $A_1, \cdots, A_k$ of every round.

---

**Algorithm 1** FixedParamComp$(\mathcal{A}, \mathcal{E} = (\mathcal{E}_1, \cdots, \mathcal{E}_k), b)$, where $\mathcal{A}$ is a randomized algorithm, $\mathcal{E}_1, \cdots, \mathcal{E}_k$ are classes of randomized algorithms, and $b \in \{0, 1\}$.

---

    Select coin tosses $R_{\mathcal{A}}^b$ for $\mathcal{A}$ uniformly at random.
    **for** $i = 1, \cdots, k$ **do**
        $\mathcal{A} = \mathcal{A}(R_{\mathcal{A}}^b, A_1^b, \cdots, A_{i-1}^b)$ gives neighboring datasets $\mathbf{x}^{i,0}, \mathbf{x}^{i,1}$, and $\mathcal{M}_i \in \mathcal{E}_i$
        $\mathcal{A}$ receives $A_i^b = \mathcal{M}_i(\mathbf{x}^{i,b})$
    **return**   view $V^b = (R_{\mathcal{A}}^b, A_1^b, \cdots, A_k^b)$

---

**Definition 2.8** (Adaptive Composition [DRV10], [MV16])**.** We say that the sequence of parameters $\varepsilon_1, \cdots, \varepsilon_k \geq 0, \delta_1, \cdots, \delta_k \in [0, 1)$ satisfies $(\varepsilon_g, \delta_g)$-differential privacy under adaptive composition if for every adversary $\mathcal{A}$, and $\mathcal{E} = (\mathcal{E}_1, \cdots, \mathcal{E}_k)$ where $\mathcal{E}_i$ is the class of $(\varepsilon_i, \delta_i)$-DP algorithms, we have FixedParamComp$(\mathcal{A}, \mathcal{E}, \cdot)$ is $(\varepsilon_g, \delta_g)$-DP in its last argument, i.e. $V^0 \approx_{\varepsilon_g, \delta_g} V^1$.

We first state a basic composition theorem which shows that the adaptive composition satisfies differential privacy where "the parameters just add up."

**Theorem 2.9** (Basic Composition [DMNS06], [DKM$^+$06])**.** *The sequence* $\varepsilon_1, \cdots, \varepsilon_k$ *and* $\delta_1, \cdots \delta_k$ *satisfies* $(\varepsilon_g, \delta_g)$-*differential privacy under adaptive composition where* $\varepsilon_g = \sum_{i=1}^k \varepsilon_i$, *and* $\delta_g = \sum_{i=1}^k \delta_i$.

We now state the advanced composition bound from [DRV10] which gives a quadratic improvement to the basic composition bound.

**Theorem 2.10** (Advanced Composition)**.** *For any* $\hat{\delta} > 0$, *the sequence* $\varepsilon_1, \cdots, \varepsilon_k$ *and* $\delta_1, \cdots \delta_k$ *where* $\varepsilon = \varepsilon_i$ *and* $\delta = \delta_i$ *for all* $i \in [k]$ *satisfies* $(\varepsilon_g, \delta_g)$-*differential privacy under adaptive composition where* $\varepsilon_g = \varepsilon (e^\varepsilon - 1) k + \varepsilon \sqrt{2k \log(1/\hat{\delta})}$, *and* $\delta_g = k\delta + \hat{\delta}$.

This theorem can be easily generalized to hold for values of $\varepsilon_i$ that are not all equal (as done in [KOV15]). However, this is not as all-encompassing as it would appear at first blush, because this straightforward generalization would not allow for the *values* of $\varepsilon_i$ and $\delta_i$ to be chosen adaptively by the data analyst. Indeed, the definition of differential privacy itself (Definition 2.4) does not straightforwardly extend to this case. The remainder of this paper is devoted to laying out a framework for sensibly talking about the privacy parameters $\varepsilon_i$ and $\delta_i$ being chosen adaptively by the data analyst, and to prove composition theorems (including an analogue of Theorem 2.10) in this model.

## 3   Composition with Adaptively Chosen Parameters

We now introduce the model of composition with adaptive parameter selection, and define privacy in this setting.

We want to model composition as in the previous section, but allow the adversary the ability to also choose the privacy parameters $(\varepsilon_i, \delta_i)$ as a function of previous rounds of interaction. We will define the view of the interaction, similar to the view in FixedParamComp, to be the tuple that includes $\mathcal{A}$'s random coin tosses $R_{\mathcal{A}}$ and the outcomes $A = (A_1, \cdots, A_k)$ of the algorithms she chose. Formally, we define an *adaptively chosen privacy parameter composition game* in Algorithm 2 which takes as input an adversary $\mathcal{A}$, a number of rounds of interaction $k$,[7] and an experiment parameter $b \in \{0, 1\}$.

We then define the privacy loss with respect to AdaptParamComp$(\mathcal{A}, k, b)$ in the following way for a fixed view $v = (r, a)$ where $r$ represents the random coin tosses of $\mathcal{A}$ and we write $v_{<i} =$

**Algorithm 2** AdaptParamComp($\mathcal{A}, k, b$)

Select coin tosses $R_{\mathcal{A}}^b$ for $\mathcal{A}$ uniformly at random.
**for** $i = 1, \cdots, k$ **do**
    $\mathcal{A} = \mathcal{A}(R_{\mathcal{A}}^b, A_1^b, \cdots, A_{i-1}^b)$ gives neighboring $\mathbf{x}^{i,0}, \mathbf{x}^{i,1}$, parameters $(\varepsilon_i, \delta_i)$, $\mathcal{M}_i$ that is $(\varepsilon_i, \delta_i)$-DP
    $\mathcal{A}$ receives $A_i^b = \mathcal{M}_i(\mathbf{x}^{i,b})$
**return** view $V^b = (R_{\mathcal{A}}^b, A_1^b, \cdots, A_k^b)$

---

$(r, a_1, \cdots, a_{i-1})$:

$$\texttt{Loss}(v) = \log\left(\frac{\mathbb{P}\left[V^0 = v\right]}{\mathbb{P}\left[V^1 = v\right]}\right) = \sum_{i=1}^{k} \log\left(\frac{\mathbb{P}\left[\mathcal{M}_i(\mathbf{x}^{i,0}) = v_i | v_{<i}\right]}{\mathbb{P}\left[\mathcal{M}_i(\mathbf{x}^{i,1}) = v_i | v_{<i}\right]}\right) \stackrel{\text{def}}{=} \sum_{i=1}^{k} \texttt{Loss}_i(v_{\leq i}). \quad (1)$$

Note that the privacy parameters $(\varepsilon_i, \delta_i)$ depend on the previous outcomes that $\mathcal{A}$ receives. We will frequently shorten our notation $\varepsilon_t = \varepsilon_t(v_{<t})$ and $\delta_t = \delta_t(v_{<t})$ when the outcome is understood.

It no longer makes sense to claim that the privacy loss of the adaptive parameter composition experiment is bounded by any fixed constant, because the privacy parameters (with which we would presumably want to use to bound the privacy loss) are themselves random variables. Instead, we define two objects which can be used by a data analyst to control the privacy loss of an adaptive composition of algorithms.

The first object, which we call a *privacy odometer* will be parameterized by one global parameter $\delta_g$ and will provide a running real valued output that will, with probability $1 - \delta_g$, upper bound the privacy loss at each round of any adaptive composition in terms of the *realized* values of $\varepsilon_i$ and $\delta_i$ selected at each round.

**Definition 3.1** (Privacy Odometer). A function $\texttt{COMP}_{\delta_g} : \mathbb{R}_{\geq 0}^{2k} \to \mathbb{R} \cup \{\infty\}$ is a *valid privacy odometer* if for all adversaries in AdaptParamComp($\mathcal{A}, k, b$), with probability at most $\delta_g$ over $v \sim V^0$: $|\texttt{Loss}(v)| > \texttt{COMP}_{\delta_g}(\varepsilon_1, \delta_1, \cdots, \varepsilon_k, \delta_k)$.

The second object, which we call a *privacy filter*, is a stopping time rule. It takes two global parameters $(\varepsilon_g, \delta_g)$ and will at each round either output CONT or HALT. Its guarantee is that with probability $1 - \delta_g$, it will output HALT if the privacy loss has exceeded $\varepsilon_g$.

**Definition 3.2** (Privacy Filter). A function $\texttt{COMP}_{\varepsilon_g, \delta_g} : \mathbb{R}_{\geq 0}^{2k} \to \{\texttt{HALT}, \texttt{CONT}\}$ is a *valid privacy filter* for $\varepsilon_g, \delta_g \geq 0$ if for all adversaries $\mathcal{A}$ in AdaptParamComp($\mathcal{A}, k, b$), the following "bad event" occurs with probability at most $\delta_g$ when $v \sim V^0$: $|\texttt{Loss}(v)| > \varepsilon_g$ and $\texttt{COMP}_{\varepsilon_g, \delta_g}(\varepsilon_1, \delta_1, \cdots, \varepsilon_k, \delta_k) = \texttt{CONT}$.

We note two things about the usage of these objects. First, a valid privacy odometer can be used to provide a running upper bound on the privacy loss at each intermediate round: the privacy loss at round $k' < k$ must with high probability be upper bounded by $\texttt{COMP}_{\delta_g}(\varepsilon_1, \delta_1, \ldots, \varepsilon_{k'}, \delta_{k'}, 0, 0, \ldots, 0, 0)$ – i.e. the bound that results by setting all future privacy parameters to 0. This is because setting all future privacy parameters to zero is equivalent to stopping the computation at round $k'$, and is a feasible choice for the adaptive adversary $\mathcal{A}$. Second, a privacy filter can be used to guarantee that with high probability, the stated privacy budget $\varepsilon_g$ is never exceeded – the data analyst at each round $k'$ simply queries $\texttt{COMP}_{\varepsilon_g, \delta_g}(\varepsilon_1, \delta_1, \ldots, \varepsilon_{k'}, \delta_{k'}, 0, 0, \ldots, 0, 0)$ *before* she runs algorithm $k'$, and runs it only if the filter returns CONT. Again, this is guaranteed because the continuation is a feasible choice of the adversary, and the guarantees of both a filter and an odometer are quantified over all adversaries.

We first give an adaptive parameter version of the basic composition in Theorem 2.9. See the full version for the proof.

**Theorem 3.3.** *For each nonnegative $\delta_g$, $\texttt{COMP}_{\delta_g}$ is a valid privacy odometer where $\texttt{COMP}_{\delta_g}(\varepsilon_1, \delta_1, \cdots, \varepsilon_k, \delta_k) = \infty$ if $\sum_{i=1}^{k} \delta_i > \delta_g$ and otherwise $\texttt{COMP}_{\delta_g}(\varepsilon_1, \delta_1, \cdots, \varepsilon_k, \delta_k) = \sum_{i=1}^{k} \varepsilon_i$. Additionally, for any $\varepsilon_g, \delta_g \geq 0$, $\texttt{COMP}_{\varepsilon_g, \delta_g}$ is a valid privacy filter where $\texttt{COMP}_{\varepsilon_g, \delta_g}(\varepsilon_1, \delta_1, \cdots, \varepsilon_k, \delta_k) = \texttt{HALT}$ if $\sum_{t=1}^{k} \delta_t > \delta_g$ or $\sum_{i=1}^{k} \varepsilon_i > \varepsilon_g$ and CONT otherwise.*

## 4 Concentration Preliminaries

We give a useful concentration bound that will be pivotal in proving an improved valid privacy odometer and filter from that given in Theorem 3.3. To set this up, we present some notation: let $(\Omega, \mathcal{F}, \mathbb{P})$ be a probability triple where $\emptyset = \mathcal{F}_0 \subseteq \mathcal{F}_1 \subseteq \cdots \subseteq \mathcal{F}$ is an increasing sequence of $\sigma$-algebras. Let $X_i$ be a real-valued $\mathcal{F}_i$-measurable random variable, such that $\mathbb{E}[X_i | \mathcal{F}_{i-1}] = 0$ a.s. for each $i$. We then consider the martingale where $M_0 = 0$ and $M_k = \sum_{i=1}^{k} X_i, \quad \forall k \geq 1$. We use results from [dlPKLL04] and [vdG02] to prove the following (see supplementary file).

**Theorem 4.1.** *For $M_k$ given above, if there exists two random variables $C_i < D_i$ which are $\mathcal{F}_{i-1}$ measurable for $i \geq 1$ such that $C_i \leq X_i \leq D_i$ almost surely $\forall i \geq 1$. and we define $U_0^2 = 0$, and $U_k^2 = \sum_{i=1}^{k} (D_i - C_i)^2, \forall k \geq 1$, then for any fixed $k \geq 1$, $\beta > 0$ and $\delta \leq 1/e$, we have*

$$\mathbb{P}\left[|M_k| \geq \sqrt{\left(\frac{U_k^2}{4} + \beta\right)\left(2 + \log\left(\frac{U_k^2}{4\beta} + 1\right)\right)\log(1/\delta)}\right] \leq \delta.$$

We will use this martingale inequality in our analysis for deriving composition bounds for both privacy filters and odometers. The martingale we form will be the sum of the privacy loss from a sequence of randomized response algorithms from Definition 2.6. Note that for pure-differential privacy (where $\delta_i = 0$) the privacy loss at round $i$ is then $\pm \varepsilon_i$, which are fixed given the previous outcomes. See the supplementary file for the case when $\delta_i > 0$ at each round $i$.

We then use the result from Theorem 2.7 to conclude that every differentially private algorithm is a post processing function of randomized response. Thus determining a high probability bound on the martingale formed from the sum of the privacy losses of a sequence of randomized response algorithms suffices for computing a valid privacy filter or odometer.

## 5 Advanced Composition for Privacy Filters

We next show that we can essentially get the same asymptotic bound as Theorem 2.10 for the privacy filter setting using the bound in Theorem 4.1 for the martingale based on the sum of privacy losses from a sequence of randomized response algorithms (see the supplementary file for more details).

**Theorem 5.1.** $\text{COMP}_{\varepsilon_g, \delta_g}$ *is a valid privacy filter for $\delta_g \in (0, 1/e)$ and $\varepsilon_g > 0$ where* $\text{COMP}_{\varepsilon_g, \delta_g}(\varepsilon_1, \delta_1, \cdots, \varepsilon_k, \delta_k) = \text{HALT}$ *if $\sum_{i=1}^{k} \delta_i > \delta_g/2$ or if $\varepsilon_g$ is smaller than*

$$\sum_{j=1}^{k} \varepsilon_j \left(e^{\varepsilon_j} - 1\right)/2$$

$$+ \sqrt{2\left(\sum_{i=1}^{k} \varepsilon_i^2 + \frac{\varepsilon_g^2}{\log(1/\delta_g)}\right)\left(1 + \frac{1}{2}\log\left(\frac{\log(1/\delta_g)\sum_{i=1}^{k} \varepsilon_i^2}{\varepsilon_g^2} + 1\right)\right)\log(2/\delta_g)} \quad (2)$$

*and otherwise* $\text{COMP}_{\varepsilon_g, \delta_g}(\varepsilon_1, \delta_1, \cdots, \varepsilon_k, \delta_k) = \text{CONT}.$

Note that if we have $\sum_{i=1}^{k} \varepsilon_i^2 = O(1/\log(1/\delta_g))$ and set $\varepsilon_g = \Theta\left(\sqrt{\sum_{i=1}^{k} \varepsilon_i^2 \log(1/\delta_g)}\right)$ in (2), we are then getting the same asymptotic bound on the privacy loss as in [KOV15] and in Theorem 2.10 for the case when $\varepsilon_i = \varepsilon$ for $i \in [k]$. If $k\varepsilon^2 \leq \frac{1}{8\log(1/\delta_g)}$, then Theorem 2.10 gives a bound on the privacy loss of $\varepsilon\sqrt{8k\log(1/\delta_g)}$. Note that there may be better choices for the constant 28.04 that we divide $\varepsilon_g^2$ by in (2), but for the case when $\varepsilon_g = \varepsilon\sqrt{8k\log(1/\delta_g)}$ and $\varepsilon_i = \varepsilon$ for every $i \in [n]$, it is nearly optimal.

## 6 Advanced Composition for Privacy Odometers

One might hope to achieve the same sort of bound on the privacy loss from Theorem 2.10 when the privacy parameters may be chosen adversarially. However we show that this cannot be the case for any valid privacy odometer. In particular, even if an adversary selects the same privacy parameter $\varepsilon = o(\sqrt{\log(\log(n)/\delta_g)/k})$ each round but can adaptively select a time to stop interacting

with `AdaptParamComp` (which is a restricted special case of the power of the general adversary – stopping is equivalent to setting all future $\varepsilon_i, \delta_i = 0$), then we show that there can be no valid privacy odometer achieving a bound of $o(\varepsilon\sqrt{k \log{(\log(n)/\delta_g)}})$. This gives a separation between the achievable bounds for a valid privacy odometers and filters. But for privacy applications, it is worth noting that $\delta_g$ is typically set to be (much) smaller than $1/n$, in which case this gap disappears (since $\log(\log(n)/\delta_g) = (1 + o(1))\log(1/\delta_g)$ ). We prove the following with an anti-concentration bound for random walks from [LT91] (see full version).

**Theorem 6.1.** *For any $\delta_g \in (0, O(1))$ there is no valid $\mathtt{COMP}_{\delta_g}$ privacy odometer where*

$$\mathtt{COMP}_{\delta_g}(\varepsilon_1, 0, \cdots, \varepsilon_k, 0) = \sum_{i=1}^{k} \varepsilon_i \left( \frac{e^{\varepsilon_i}-1}{e^{\varepsilon_i}+1} \right) + o\left( \sqrt{\sum_{i=1}^{k} \varepsilon_i^2 \log(\log(n)/\delta_g)} \right)$$

We now give our main positive result for privacy odometers, which is similar to our privacy filter in Theorem 5.1 except that $\delta_g$ is replaced by $\delta_g/\log(n)$, as is necessary from Theorem 6.1. Note that the bound incurs an additive $1/n^2$ loss to the $\sum_i \varepsilon_i^2$ term that is present without privacy. In any reasonable setting of parameters, this translates to at most a constant-factor multiplicative loss, because there is no utility running any differentially private algorithm with $\varepsilon_i < \frac{1}{10n}$ (we know that if $A$ is $(\varepsilon_i, 0)$-DP then $A(\mathbf{x})$ and $A(\mathbf{x}')$ for neighboring inputs have statistical distance at most $e^{\varepsilon_i n} - 1 < 0.1$, and hence the output is essentially independent of the input - note that a similar statement holds for $(\varepsilon_i, \delta_i)$-DP.) The proof of the following result uses Theorem 4.1 along with a union bound over $\log(n^2)$ choices for $\beta$, which are discretized values for $\sum_{i=1}^{k} \varepsilon_i^2 \in [1/n^2, 1]$. See the full version for the complete proof.

**Theorem 6.2** (Advanced Privacy Odometer). $\mathtt{COMP}_{\delta_g}$ *is a valid privacy odometer for $\delta_g \in (0, 1/e)$ where $\mathtt{COMP}_{\delta_g}(\varepsilon_1, \delta_1, \cdots, \varepsilon_k, \delta_k) = \infty$ if $\sum_{i=1}^{k} \delta_i > \delta_g/2$, otherwise if $\sum_{i=1}^{k} \varepsilon_i^2 \in [1/n^2, 1]$ then*

$$\mathtt{COMP}_{\delta_g}(\varepsilon_1, \delta_1, \cdots, \varepsilon_k, \delta_k) = \sum_{i=1}^{k} \varepsilon_i \left( \frac{e^{\varepsilon_i}-1}{2} \right) + 2\sqrt{\sum_{i=1}^{k} \varepsilon_i^2 \left( 1 + \log\left(\sqrt{3}\right) \right) \log(4\log_2(n)/\delta_g)}.$$

(3)

*and if $\sum_{i=1}^{k} \varepsilon_i^2 \notin [1/n^2, 1]$ then $\mathtt{COMP}_{\delta_g}(\varepsilon_1, \delta_1, \cdots, \varepsilon_k, \delta_k)$ is equal to*

$$\sum_{i=1}^{k} \varepsilon_i \left( \frac{e^{\varepsilon_i}-1}{2} \right) + \sqrt{2\left( 1/n^2 + \sum_{i=1}^{k} \varepsilon_i^2 \right) \left( 1 + \frac{1}{2}\log\left( 1 + n^2\sum_{i=1}^{k} \varepsilon_i^2 \right) \right) \log(4\log_2(n))/\delta_g}.$$

(4)

## Acknowledgements

The authors are grateful Jack Murtagh for his collaboration in the early stages of this work, and for sharing his preliminary results with us. We thank Andreas Haeberlen, Benjamin Pierce, and Daniel Winograd-Cort for helpful discussions about composition. We further thank Daniel Winograd-Cort for catching an incorrectly set constant in an earlier version of Theorem 5.1.

## Footnotes

[5]A *statistical query* is parameterized by a predicate $\phi$, and asks "how many elements of the dataset satisfy $\phi$?" Changing a single element of the dataset can change the answer to the statistical query by at most 1.

[6]The same analysis holds for hetereogeneous parameters $(\varepsilon_1, \dots, \varepsilon_k)$ are used in each round as long as they are all fixed in advance. For basic composition $\varepsilon k$ is replaced with $\sum_{i=1}^{k} \varepsilon_i$ and for advanced composition $\varepsilon\sqrt{k}$ is replaced with $\sqrt{\sum_{i=1}^{k} \varepsilon_i^2}$.

[7]Note that in the adaptive parameter composition game, the adversary has the option of effectively stopping the composition early at some round $k' < k$ by simply setting $\varepsilon_i = \delta_i = 0$ for all rounds $i > k'$. Hence, the parameter $k$ will not appear in our composition theorems the way it does when privacy parameters are fixed. This means that we can effectively take $k$ to be infinite. For technical reasons, it is simpler to have a finite parameter $k$, but the reader should imagine it as being an enormous number.

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
