[Supplementary Material]

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

 give the formal description of this interaction where $\mathcal{A}$ uses the privacy filter in Algorithm 3.

From the way we have defined a valid privacy filter, we have the following proposition:

**Proposition 3.3.** *If* $\mathrm{COMP}_{\varepsilon_g, \delta_g}$ *is a valid privacy filter then the views* $V^0_{\mathrm{F}}$ *and* $V^1_{\mathrm{F}}$ *of the adversary from* $\mathtt{PrivacyFilterComp}\left(\mathcal{A}, k, b; \mathrm{COMP}_{\varepsilon_g, \delta_g}\right)$ *with* $b = 0$ *and* $b = 1$ *respectively, are* $(\varepsilon_g, \delta_g)$-*point-wise indistinguishable and hence* $V^0_{\mathrm{F}} \approx_{\varepsilon_g, \delta_g} V^1_{\mathrm{F}}$.

## 3.2 Focusing on Randomized Response

In [KOV15, MV16] Theorem 2.7 was used to show that for ordinary composition (Theorem 2.8), it suffices to analyze the composition of randomized response. In this section, we show something similar for privacy odometers and filters. Specifically, we show that we can simulate

---

**Algorithm 3** PrivacyFilterComp($\mathcal{A}, k, b; \text{COMP}_{\varepsilon_g, \delta_g}$)

---

Select coin tosses $R_{\mathcal{A}}^b$ for $\mathcal{A}$ uniformly at random.
**for** $i = 1, \cdots, k$ **do**
$\quad \mathcal{A} = \mathcal{A}(R_{\mathcal{A}}^b, A_1^b, \cdots, A_{i-1}^b)$ gives neighboring $\mathbf{x}^{i,0}, \mathbf{x}^{i,1}$, $(\varepsilon_i, \delta_i)$, and $\mathcal{M}_i$ that is $(\varepsilon_i, \delta_i)$-DP
$\quad$ **if** $\text{COMP}_{\varepsilon_g, \delta_g}(\varepsilon_1, \delta_1, \cdots, \varepsilon_i, \delta_i, 0, 0, \cdots, 0, 0) = \text{HALT}$ **then**
$\quad\quad A_i, \cdots, A_k = \perp$
$\quad\quad$ BREAK
$\quad$ **else**
$\quad\quad \mathcal{A}$ receives $A_i^b = \mathcal{M}_i(\mathbf{x}^{i,b})$
**return** view $V_{\text{F}}^b = (R_{\mathcal{A}}^b, A_1^b, \cdots, A_k^b)$

---

AdaptParamComp($\mathcal{A}, k, b$) by defining a new adversary that chooses the differentially private algorithm $\mathcal{M}_i$ of adversary $\mathcal{A}$, but uses the randomized response algorithm from Theorem 2.6 each round along with a post-processing function, which together determine the distribution for $\mathcal{M}_i$.

In Algorithm 4, we define the new composition game SimulatedComp($\mathcal{A}, k, b$) with adversary $\mathcal{A}$ that outputs the view $W^b$, which includes the internal randomness $R_{\mathcal{A}}^b$ of $\mathcal{A}$ with the randomized response outcomes $Z^b = (Z_1^b, \cdots, Z_k^b)$. From Theorem 2.7, we know that we can simulate any $(\varepsilon, \delta)$-DP algorithm as a randomized post-processing function $T$ on top of $\text{RR}_{\varepsilon, \delta}$. Thus given the outcomes prior to round $i$, $\mathcal{A}$ selects $\mathcal{M}_i$, which is equivalent to selecting a post-processing function $T_i$. Note that we can simulate $T_i$ as a deterministic function $P_i$ with access to random coins $R_{\text{SIM}_i}^b$, i.e. $P_i\left(\text{RR}_{\varepsilon_i, \delta_i}(b); R_{\text{SIM}_i}^b\right) \sim T_i\left(\text{RR}_{\varepsilon_i, \delta_i}(b)\right)$. We then include the random coins $R_{\text{SIM}}^b = \left(R_{\text{SIM}_1}^b, \cdots, R_{\text{SIM}_k}^b\right)$ in the view of adversary $\mathcal{A}$ in SimulatedComp($\mathcal{A}, k, b$). From the view $W^b = \left(R_{\mathcal{A}}^b, R_{\text{SIM}}, Z_1^b, \cdots, Z_k^b\right)$, $\mathcal{A}$ would be able to reconstruct the privacy parameters selected each round along with algorithms $\mathcal{M}_1, \cdots, \mathcal{M}_k$ used, which would also determine the post-processing functions $P_1, \cdots, P_k$.

---

**Algorithm 4** SimulatedComp($\mathcal{A}, k, b$)

---

Select coin tosses $R_{\mathcal{A}}^b$ for $\mathcal{A}$ uniformly at random.
**for** $i = 1, \cdots, k$ **do**
$\quad \mathcal{A} = \mathcal{A}\left(R_{\mathcal{A}}^b, Y_1^b, \cdots, Y_{i-1}^b\right)$ gives neighboring $\mathbf{x}^{i,0}, \mathbf{x}^{i,1}$, parameters $(\varepsilon_i, \delta_i)$, $\mathcal{M}_i$ that is $(\varepsilon_i, \delta_i)$-DP.

$\quad$ Let $P_i$ be a deterministic post-processing function, such that

$$P_i\left(\text{RR}_{\varepsilon_i, \delta_i}(b); R_{\text{SIM}_i}^b\right) \sim \mathcal{M}_i\left(\mathbf{x}^{i,b}\right) \tag{2}$$

$\quad$ for uniformly random $R_{\text{SIM}_i}^b$.
$\quad$ Compute $Z_i^b = \text{RR}_{\varepsilon_i, \delta_i}(b)$ and $Y_i^b = P_i(Z_i^b; R_{\text{SIM}_i}^b)$.
$\quad \mathcal{A}$ receives $Y_i^b$.
**return** view $W^b = (R_{\mathcal{A}}^b, R_{\text{SIM}}^b, Z_1^b, \cdots, Z_k^b)$, where $R_{\text{SIM}}^b = \left(R_{\text{SIM}_1}^b, \cdots, R_{\text{SIM}_k}^b\right)$.

---

From the way that we have defined $P_i$ in (2), for each fixed value of the internal randomness of $\mathcal{A}$, the view of $\mathtt{AdaptParamComp}(\mathcal{A}, k, b)$ is distributed identically to a post-processing of the view $W^b$ from $\mathtt{SimulatedComp}(\mathcal{A}, k, b)$.

**Lemma 3.4.** *For every adversary $\mathcal{A}$, the deterministic function $P$ defined as*

$$P\left(R_{\mathcal{A}}^b, R_{\mathtt{SIM}}^b, Z_1^b, \cdots, Z_k^b\right) = \left(R_{\mathcal{A}}^b, P_1(Z_1^b; R_{\mathtt{SIM}_1}^b), \cdots, P_k(Z_k^b; R_{\mathtt{SIM}_k}^b)\right) \tag{3}$$

*ensures $P\left(\mathtt{SimulatedComp}(\mathcal{A}, k, b)\right)$ and $\mathtt{AdaptParamComp}(\mathcal{A}, k, b)$ are identically distributed.*

Since $R_{\mathcal{A}}^b$ is the first argument of both random variables, they are also identically distributed conditioned on any fixed value of $R_{\mathcal{A}}^b$. This point-wise equivalence for every value of the internal randomness allows us to without loss of generality analyze *deterministic* adversaries and post-processing functions of $\mathtt{SimulatedComp}(\mathcal{A}, k, b)$ in order to reason about the view of $\mathtt{AdaptParamComp}(\mathcal{A}, k, b)$. Because the randomness is fixed, for clarity, we will omit the random coins $R_{\mathcal{A}}^b$ from the view of both composition games for the rest of the analysis.

We will now show that it is sufficient to prove bounds in which $\varepsilon_i$ may be adaptively chosen at each round, and in which $\{\delta_i\} \equiv 0$ uniformly. We do this by giving a generic way to extend a bound in the $\delta_i = 0$ case to a bound that holds when the $\delta_i$ may be non-zero. Define a slight modification of Algorithm 4 called $\widetilde{\mathtt{SimulatedComp}}(\mathcal{A}, k, b)$ which is the same as $\mathtt{SimulatedComp}(\mathcal{A}, k, b)$ except that it computes $\widetilde{Z}_i^b = \mathtt{RR}_{\varepsilon_i}(b)$ (where $\delta_i = 0$) and sets $\widetilde{Y}_i^b = P_i\left(\widetilde{Z}_i^b; R_{\mathtt{SIM}_i}^b\right)$. We then define the final view of the adversary $\mathcal{A}$ in $\widetilde{\mathtt{SimulatedComp}}(\mathcal{A}, k, b)$ as $\widetilde{W}^b$ where

$$\widetilde{W}^b = \left(R_{\mathtt{SIM}}^b, \widetilde{Z}_1^b, \cdots, \widetilde{Z}_k^b\right) \qquad \text{and} \qquad \widetilde{V}^b = \left(\widetilde{Y}_1^b, \cdots, \widetilde{Y}_k^b\right) = P\left(\widetilde{W}^b\right) \tag{4}$$

for $P(\cdot)$ given in (3). We then say that $\widetilde{\mathtt{COMP}}_{\delta_g}$ (also $\widetilde{\mathtt{COMP}}_{\varepsilon_g, \delta_g}$) is a valid privacy odometer (filter) when $\{\delta_i\} \equiv 0$ if over all deterministic adversaries $\mathcal{A}$ in $\widetilde{\mathtt{SimulatedComp}}(\mathcal{A}, k, b)$ the condition in Theorem 3.1 (Theorem 3.2) holds with probability at most $\delta_g$ over $\widetilde{v} \sim P\left(\widetilde{W}^b\right)$ except now the privacy loss is given as

$$\widetilde{\mathtt{Loss}}(\widetilde{v}) = \log\left(\frac{\mathbb{P}\left[\widetilde{V}^0 = \widetilde{v}\right]}{\mathbb{P}\left[\widetilde{V}^1 = \widetilde{v}\right]}\right) = \sum_{i=1}^k \log\left(\frac{\mathbb{P}\left[P_i\left(\mathtt{RR}_{\varepsilon_i}(0); R_{\mathtt{SIM}_i}^0\right) = \widetilde{v}_i | \widetilde{v}_{<i}\right]}{\mathbb{P}\left[P_i\left(\mathtt{RR}_{\varepsilon_i}(1); R_{\mathtt{SIM}_i}^1\right) = \widetilde{v}_i | \widetilde{v}_{<i}\right]}\right) \stackrel{\text{def}}{=} \sum_{i=1}^k \widetilde{\mathtt{Loss}}_i(\widetilde{v}_{\leq i}). \tag{5}$$

The following result gives the connection between valid privacy odometers and filters in the modified game $\widetilde{\mathtt{SimulatedComp}}(\mathcal{A}, k, b)$ with the original definitions given in Theorems 3.1 and 3.2.

**Lemma 3.5.** *If $\widetilde{\mathtt{COMP}}_{\delta_g}$ is a valid privacy odometer when $\{\delta_i\} \equiv 0$, then for every $\delta_g' \geq 0$, $\mathtt{COMP}_{\delta_g + \delta_g'}$ is a valid privacy odometer where*

$$\mathtt{COMP}_{\delta_g + \delta_g'}\left(\varepsilon_1, \delta_1, \cdots, \varepsilon_k, \delta_k\right) = \begin{cases} \infty & \text{if } \sum_{i=1}^k \delta_i > \delta_g' \\ \widetilde{\mathtt{COMP}}_{\delta_g}\left(\varepsilon_1, 0, \cdots, \varepsilon_k, 0\right) & \text{otherwise} \end{cases}.$$

*If $\widetilde{\mathtt{COMP}}_{\varepsilon_g, \delta_g}$ is a valid privacy filter when $\{\delta_i\} \equiv 0$, then for every $\delta_g' \geq 0$, $\mathtt{COMP}_{\varepsilon_g, \delta_g + \delta_g'}$ is a valid privacy filter where*

$$\mathtt{COMP}_{\varepsilon_g, \delta_g + \delta_g'}\left(\varepsilon_1, \delta_1, \cdots, \varepsilon_k, \delta_k\right) = \begin{cases} \mathtt{HALT} & \text{if } \sum_{i=1}^k \delta_i > \delta_g' \\ \widetilde{\mathtt{COMP}}_{\varepsilon_g, \delta_g}\left(\varepsilon_1, 0, \cdots, \varepsilon_k, 0\right) & \text{otherwise} \end{cases}.$$

*Proof.* Let $W = (R_{\text{SIM}}, Z_1, \cdots, Z_k)$ be the view of $\mathcal{A}$ in $\texttt{SimulatedComp}(\mathcal{A}, k, 0)$ and $\widetilde{W} = (R_{\text{SIM}}, \widetilde{Z}_1, \cdots, \widetilde{Z}_k)$ be her view in $\widetilde{\texttt{SimulatedComp}}(A, k, 0)$ (where $\{\delta_i\} \equiv 0$). We will also write the view of $\texttt{AdaptParamComp}(\mathcal{A}, k, 0)$ as $V = (A_1, \cdots, A_k)$ and the post-processing functions of $\mathcal{A}$ as $P_i$ from (2). As in (3), we will use the notation $P(W) = \big(P_1(Z_1; R_{\text{SIM}_i}), \cdots, P_k(Z_k; R_{\text{SIM}_i})\big)$ and similarly for $\widetilde{V} = P(\widetilde{W})$. Recall that from Theorem 3.4 that we know $V \sim P(W)$, even if $\mathcal{A}$ were randomized.

Consider the following method of sampling from $\text{RR}_{\varepsilon, \delta}$: first select outcome $\widetilde{z}$ from $\text{RR}_\varepsilon(0)$, then with probability $1 - \delta$ set $z = \widetilde{z}$ – otherwise set $z = 0$. Note that this samples from the correct distribution for $\text{RR}_{\varepsilon, \delta}(0)$. We can thus couple draws from $\text{RR}_\varepsilon(0)$ and $\text{RR}_{\varepsilon, \delta}(0)$, so for our setting we write the coupled random variable as: $\mathbf{V} = (V, \widetilde{V})$.

We then define the following sets:

$$\mathcal{F} = \{(w = (r, z), \widetilde{w} = (r, \widetilde{z})) : \exists t \in [k] \text{ s.t. } z_t \neq \widetilde{z}_t\}, \quad \mathcal{G}_t = \left\{v : \sum_{i=1}^t \delta_i(v_{<i}) \leq \delta'_g\right\},$$

$$\mathcal{F}_t = \{(w = (r, z), \widetilde{w} = (r, \widetilde{z})) : z_t \neq \widetilde{z}_t \text{ and } z_i = \widetilde{z}_i \quad \forall i < t\}, \quad \mathcal{H} = \left\{v : |\widetilde{\text{Loss}}(v)| \geq \widetilde{\text{COMP}}_{\delta_g}(\varepsilon_1, 0, \cdots, \varepsilon_k, 0)\right\}.$$

We then want to show that we can bound the privacy loss with high probability. Specifically,

$$\mathbb{P}_{\mathbf{V}}\left[|\text{Loss}(V)| \geq \widetilde{\text{COMP}}_{\delta_g}(\varepsilon_1, 0, \cdots, \varepsilon_k, 0) \quad \wedge \quad \sum_{t=1}^k \delta_t \leq \delta'_g\right] \leq \delta_g + \delta'_g.$$

where each $\varepsilon_i$ is a function of the outputs of the prefix $\widetilde{V}_{<i}$ of the full view $\widetilde{V}$ from $\widetilde{\texttt{SimulatedComp}}(\mathcal{A}, k, 0)$. We now show that the quantity that we want to bound can be written as the probability of the coupled random variables $\mathbf{V}$ and $\mathbf{W} = \big(W, \widetilde{W}\big)$ being contained in the sets that we defined above.

$$\mathbb{P}_{\mathbf{V} \sim \big(P(W), P(\widetilde{W})\big)}\left[|\text{Loss}(V)| \geq \widetilde{\text{COMP}}_{\delta_g}(\varepsilon_1, 0, \cdots, \varepsilon_k, 0) \quad \wedge \quad \sum_{t=1}^k \delta_t \leq \delta'_g\right]$$

$$\leq \mathbb{P}\left[(\mathbf{W} \in \mathcal{F} \quad \wedge \quad V \in \mathcal{G}_k) \quad \vee \quad (V \in \mathcal{H} \quad \wedge \quad \mathbf{W} \notin \mathcal{F})\right]$$

$$\leq \mathbb{P}\left[\mathbf{W} \in \mathcal{F} \quad \wedge \quad V \in \mathcal{G}_k\right] + \mathbb{P}\left[\widetilde{V} \in \mathcal{H}\right]$$

$$\leq \mathbb{P}\left[\mathbf{W} \in \mathcal{F} \quad \wedge \quad V \in \mathcal{G}_k\right] + \delta_g \tag{6}$$

Note, that if $\sum_{i=1}^k \delta_i(v_{<i}) \leq \delta_g$ then we must have $\sum_{i=1}^t \delta_i(v_{<i}) \leq \delta_g$ for each $t < k$, so that $\mathcal{G}_k \subseteq \mathcal{G}_t$. We then use the fact that $\{\mathcal{F}_t : t \in [k]\}$ forms a partition of $\mathcal{F}$, i.e. $\mathcal{F} = \bigcup_{t=1}^k \mathcal{F}_t$ and $\mathcal{F}_i \cap \mathcal{F}_j = \emptyset$ for $i \neq j$, to obtain the following:

$$\mathbb{P}[\mathbf{W} \in \mathcal{F} \quad \wedge \quad V \in \mathcal{G}_k] = \sum_{t=1}^k \mathbb{P}[\mathbf{W} \in \mathcal{F}_t \quad \wedge \quad V \in \mathcal{G}_k] \leq \sum_{t=1}^k \mathbb{P}[\mathbf{W} \in \mathcal{F}_t \quad \wedge \quad V \in \mathcal{G}_t].$$

$$\sum_{t=1}^{k} \mathbb{P}[\mathbf{W} \in \mathcal{F}_t \quad \wedge \quad V \in \mathcal{G}_t]$$

$$\leq \sum_{t=1}^{k} \mathbb{P}\left[\mathbf{W} \in \mathcal{F}_t \quad \wedge \quad \widetilde{V} \in \mathcal{G}_t\right]$$

$$= \sum_{t=1}^{k} \sum_{\widetilde{v} \in \mathcal{G}_t} \mathbb{P}\left[\widetilde{V} = \widetilde{v}\right] \mathbb{P}[\mathbf{W} \in \mathcal{F}_t | \widetilde{v}]$$

$$\leq \sum_{t=1}^{k} \sum_{\widetilde{v} \in \mathcal{G}_t} \mathbb{P}\left[\widetilde{V} = \widetilde{v}\right] \delta_t(\widetilde{v}_{<t}).$$

We now switch the order of summation to obtain our result

$$\sum_{t=1}^{k} \sum_{\widetilde{v} \in \mathcal{G}_t} \mathbb{P}\left[\widetilde{V} = \widetilde{v}\right] \delta_t(\widetilde{v}_{<t}) = \sum_{\widetilde{v}} \mathbb{P}\left[\widetilde{V} = \widetilde{v}\right] \sum_{t : \sum_{i=1}^{t} \delta_i(\widetilde{v}_{<i}) \leq \delta'_g} \delta_t(\widetilde{v}_{<t}) \leq \sum_{\widetilde{v}} \mathbb{P}\left[\widetilde{V} = \widetilde{v}\right] \delta'_g = \delta'_g. \qquad (7)$$

We then combine this with (6) to prove our first statement for the privacy odometer.

Using the same notation as above, we now move to proving the statement for the privacy filter. It suffices to prove the following:

$$\mathbb{P}_{\mathbf{V} \sim (P(W), P(\widetilde{W}))}\left[|\mathtt{Loss}(V)| \geq \varepsilon_g \quad \wedge \quad V \in \mathcal{G}_k \quad \wedge \quad \widehat{\mathtt{COMP}}_{\varepsilon_g, \delta_g}(\varepsilon_1, 0, \cdots, \varepsilon_k, 0) = \mathtt{CONT}\right] \leq \delta_g + \delta'_g.$$

We now define a slight variant of $\mathcal{H}$ from above:

$$\mathcal{H}_{\varepsilon_g} = \left\{v : \left|\widehat{\mathtt{Loss}}(v)\right| \geq \varepsilon_g\right\}.$$

Similar to what we showed in (6) for the privacy odometer, we have

$$\mathbb{P}_{\mathbf{V}}\left[|\mathtt{Loss}(V)| \geq \varepsilon_g \quad \wedge \quad V \in \mathcal{G}_k \quad \wedge \quad \widehat{\mathtt{COMP}}_{\varepsilon_g, \delta_g}(\varepsilon_1, 0, \cdots, \varepsilon_k, 0) = \mathtt{CONT}\right]$$

$$\leq \mathbb{P}_{\mathbf{V} \sim (P(W), P(\widetilde{W}))}\left[\left((\mathbf{W} \in \mathcal{F} \wedge V \in \mathcal{G}_k) \vee (V \in \mathcal{H}_{\varepsilon_g} \wedge \mathbf{W} \notin \mathcal{F})\right) \quad \wedge \quad \widehat{\mathtt{COMP}}_{\varepsilon_g, \delta_g}(\varepsilon_1, 0, \cdots, \varepsilon_k, 0) = \mathtt{CONT}\right]$$

$$\leq \mathbb{P}[\mathbf{W} \in \mathcal{F} \quad \wedge \quad V \in \mathcal{G}_k] + \mathbb{P}\left[\widetilde{V} \in \mathcal{H}_{\varepsilon_g} \quad \wedge \quad \widehat{\mathtt{COMP}}_{\varepsilon_g, \delta_g}(\varepsilon_1, 0, \cdots, \varepsilon_k, 0) = \mathtt{CONT}\right]$$

$$\leq \mathbb{P}[\mathbf{W} \in \mathcal{F} \quad \wedge \quad V \in \mathcal{G}_k] + \delta_g$$

$$\leq \delta'_g + \delta_g$$

where the last inequality follows from (6) and (7). $\qquad \square$

## 3.3   Basic Composition

We first give an adaptive parameter version of the basic composition in Theorem 2.9.

**Theorem 3.6.** *For each nonnegative $\delta_g$, $\mathtt{COMP}_{\delta_g}$ is a valid privacy odometer where*

$$\mathtt{COMP}_{\delta_g + \delta'_g}(\varepsilon_1, \delta_1, \cdots, \varepsilon_k, \delta_k) = \begin{cases} \infty & \text{if } \sum_{i=1}^{k} \delta_i > \delta'_g \\ \sum_{i=1}^{k} \varepsilon_i & \text{otherwise} \end{cases}.$$

*Additionally, for any $\varepsilon_g, \delta_g \geq 0$, $\mathrm{COMP}_{\varepsilon_g, \delta_g}$ is a valid privacy filter where*

$$\mathrm{COMP}_{\varepsilon_g, \delta_g}(\varepsilon_1, \delta_1, \cdots, \varepsilon_k, \delta_k) = \begin{cases} \mathrm{HALT} & \text{if } \sum_{i=1}^{k} \delta_i > \delta_g' \text{ or } \sum_{i=1}^{k} \varepsilon_i > \varepsilon_g \\ \mathrm{CONT} & \text{otherwise} \end{cases}.$$

*Proof.* We use Theorems 3.4 and 3.5 so that we need to only reason about any deterministic adversary in $\mathtt{Simul\widetilde{ate}dComp}(\mathcal{A}, k, b)$. We know that $(\varepsilon, 0)$-DP is closed under post-processing from Theorem 2.5, so that for any (randomized) post-processing function $T$, we have $T(\mathrm{RR}_\varepsilon(0)) \approx_{\varepsilon, 0} T(\mathrm{RR}_\varepsilon(1))$ and by Theorem 2.3 we know that $T(\mathrm{RR}_\varepsilon(0))$ and $T(\mathrm{RR}_\varepsilon(1))$ are $(\varepsilon, 0)$-point-wise indistinguishable for any post-processing function $T$. The proof then follows simply from the definition of (pure) differential privacy, so for all possible views $\widetilde{v}$ of the adversary in $P\left(\mathtt{Simul\widetilde{ate}dComp}(\mathcal{A}, k, b)\right)$:

$$\left| \widetilde{\mathrm{Loss}}(\widetilde{v}) \right| \leq \sum_{i=1}^{k} \left| \log \left( \frac{\mathbb{P}\left[ P_i\left( \mathrm{RR}_{\varepsilon_i}(0); R_{\mathrm{SIM}_i}^0 \right) = \widetilde{v}_i | \widetilde{v}_{<i} \right]}{\mathbb{P}\left[ P_i\left( \mathrm{RR}_{\varepsilon_i}(1); R_{\mathrm{SIM}_i}^1 \right) = \widetilde{v}_i | \widetilde{v}_{<i} \right]} \right) \right| \leq \sum_{i=1}^{k} \varepsilon_i(\widetilde{v}_{<i})$$

where we explicitly write the dependence of the choice of $\varepsilon_i$ by $\mathcal{A}$ at round $i$ on the view from the previous rounds as $\varepsilon_i(\widetilde{v}_{<i})$ $\qquad\square$

# 4 Concentration Preliminaries

We give a useful concentration bound that will be pivotal in proving an improved valid privacy odometer and filter from that given in Theorem 3.6. We first present a concentration bound for *self normalized processes*.

**Lemma 4.1** (Corollary 2.2 in [dlPKLL04]). *If $A$ and $B > 0$ are two random variables such that*

$$\mathbb{E}\left[ \exp\left( \lambda A - \frac{\lambda^2}{2} B^2 \right) \right] \leq 1 \tag{8}$$

*for all $\lambda \in \mathbb{R}$, then for all $\delta \leq 1/e$, $\beta > 0$ we have*

$$\mathbb{P}\left[ |A| \geq \sqrt{(B^2 + \beta)\left( 2 + \log\left( \frac{B^2}{\beta} + 1 \right) \right) \log(1/\delta)} \right] \leq \delta.$$

To put this bound into context, suppose that $B$ is a constant and we apply the bound with $\beta = B^2$. Then the bound simplifies to

$$\mathbb{P}\left[ |A| \geq O\left( B\sqrt{\log(1/\delta)} \right) \right] \leq \delta,$$

which is just a standard concentration inequality for any subgaussian random variable $A$ with standard deviation $B$.

We will apply Theorem 4.1 to random variables coming from martingales defined from the privacy loss functions. To set this up, we present some notation: let $(\Omega, \mathcal{F}, \mathbb{P})$ be a probability triple where $\emptyset = \mathcal{F}_0 \subseteq \mathcal{F}_1 \subseteq \cdots \subseteq \mathcal{F}$ is an increasing sequence of $\sigma$-algebras. Let $X_i$ be a real-valued

$\mathcal{F}_i$-measurable random variable, such that $\mathbb{E}[X_i|\mathcal{F}_{i-1}] = 0$ a.s. for each $i$. We then consider the martingale where

$$M_0 = 0 \qquad M_k = \sum_{i=1}^{k} X_i, \qquad \forall k \geq 1. \tag{9}$$

We then use the following result which gives us a pair of random variables to which we can apply Theorem 4.1.

**Lemma 4.2** (Lemma 2.4 in [vdG02]). *For $M_k$ defined in* (9), *if there exists two random variables $C_i < D_i$ that are $\mathcal{F}_{i-1}$-measurable for $i \geq 1$*

$$C_i \leq X_i \leq D_i \qquad a.s. \quad \forall i \geq 1.$$

*and we define $U_k$ as*

$$U_0^2 = 0, \qquad U_k^2 = \sum_{i=1}^{k} (D_i - C_i)^2, \quad \forall k \geq 1 \tag{10}$$

*then*

$$\exp\left[\lambda M_k - \frac{\lambda^2}{8} U_k^2\right]$$

*is a supermartingale for all $\lambda \in \mathbb{R}$.*

We then obtain the following result from combining Theorem 4.1 with Theorem 4.2.

**Theorem 4.3.** *Let $M_k$ be defined as in* (9) *and satisfy the hypotheses of Theorem 4.2. Then for every fixed $k \geq 1$, $\beta > 0$ and $\delta \leq 1/e$, we have*

$$\mathbb{P}\left[|M_k| \geq \sqrt{\left(\frac{U_k^2}{4} + \beta\right)\left(2 + \log\left(\frac{U_k^2}{4\beta} + 1\right)\right)\log(1/\delta)}\right] \leq \delta$$

Given Theorem 3.5, we will focus on finding a valid privacy odometer and filter when $\{\delta_i\} \equiv 0$. Our analysis will then depend on the privacy loss $\widetilde{\text{Loss}}(\widetilde{V})$ from (5) where $\widetilde{V}$ is the view of the adversary in $\texttt{Simu\widetilde{late}dComp}(\mathcal{A}, k, 0)$. We then focus on the following martingale in our analysis:

$$\widetilde{M}_k = \sum_{i=1}^{k} \left(\widetilde{\text{Loss}}_i(\widetilde{V}_{\leq i}) - \widetilde{\mu}_i\right) \qquad \text{where} \qquad \widetilde{\mu}_i = \mathbb{E}\left[\widetilde{\text{Loss}}_i(\widetilde{V}_{\leq i})\big|\widetilde{V}_{<i}\right]. \tag{11}$$

We can then bound the conditional expectation $\widetilde{\mu}_i$ with the following result from [DR16] that improves on an earlier result from [DRV10] by a factor of 2.

**Lemma 4.4** ([DR16]). *For $\widetilde{\mu}_i$ defined in* (11), *we have $\widetilde{\mu}_i \leq \varepsilon_i (e^{\varepsilon_i} - 1)/2$.*

# 5  Advanced Composition for Privacy Filters

We next show that we can essentially get the same asymptotic bound as Theorem 2.10 for the privacy filter setting using the bound in Theorem 4.3 for the martingale given in (11).

**Theorem 5.1.** *We define $\mathcal{K}$ as the following*

$$\mathcal{K} \overset{def}{=} \sum_{j=1}^{k} \varepsilon_j \left( \frac{e^{\varepsilon_j} - 1}{2} \right) + \sqrt{2 \left( \sum_{i=1}^{k} \varepsilon_i^2 + \frac{\varepsilon_g^2}{28.04 \cdot \log(1/\delta_g)} \right) \left( 1 + \frac{1}{2} \log \left( \frac{28.04 \cdot \log(1/\delta_g) \sum_{i=1}^{k} \varepsilon_i^2}{\varepsilon_g^2} + 1 \right) \right) \log(2/\delta_g)}.$$

(12)

$\mathtt{COMP}_{\varepsilon_g, \delta_g}$ *is a valid privacy filter for $\delta_g \in (0, 1/e)$ and $\varepsilon_g > 0$ where*

$$\mathtt{COMP}_{\varepsilon_g, \delta_g}(\varepsilon_1, \delta_1, \cdots, \varepsilon_k, \delta_k) = \begin{cases} \mathtt{HALT} & \text{if } \sum_{i=1}^{k} \delta_i > \delta_g/2 \text{ or } \mathcal{K} > \varepsilon_g \\ \mathtt{CONT} & \text{otherwise} \end{cases}.$$

Note that if we have $\sum_{i=1}^{k} \varepsilon_i^2 = O\left(1/\log(1/\delta_g)\right)$ and set $\varepsilon_g = \Theta\left( \sqrt{\sum_{i=1}^{k} \varepsilon_i^2 \log(1/\delta_g)} \right)$ in (12), we are then getting the same asymptotic bound on the privacy loss as in [KOV15] and in Theorem 2.10 for the case when $\varepsilon_i = \varepsilon$ for $i \in [k]$. If $k\varepsilon^2 \le \frac{1}{8\log(1/\delta_g)}$, then Theorem 2.10 gives a bound on the privacy loss of $\varepsilon \sqrt{8k \log(1/\delta_g)}$. Note that there may be better choices for the constant 28.04 that we divide $\varepsilon_g^2$ by in (12), but for the case when $\varepsilon_g = \varepsilon \sqrt{8k \log(1/\delta_g)}$ and $\varepsilon_i = \varepsilon$ for every $i \in [n]$, it is nearly optimal.

*Proof of Theorem 5.1.* Note that Theorem 3.5 allows us to concentrate on showing that we can find an optimal privacy filter when $\{\delta_i\} \equiv 0$. We then focus on the martingale $\widetilde{M}_k$ given in (11). In order to apply Theorem 4.3 we set the lower bound for $\widetilde{M}_i$ to be $C_i = (-\varepsilon_i - \widetilde{\mu}_i)$ and upper bound to be $D_i = (\varepsilon_i - \widetilde{\mu}_i)$ in order to compute $U_k^2$ from (10). We then have for the martingale in (11) that

$$U_k^2 = 4 \sum_{i=1}^{k} \varepsilon_i^2.$$

We can then directly apply Theorem 4.3 to get the following for $\beta = \left( \frac{\varepsilon_g}{\sqrt{28.04 \cdot \log(1/\delta_g)}} \right)^2 > 0$ with probability at least $1 - \delta_g/2$

$$|\widetilde{M}_k| \le \sqrt{2 \left( \sum_{i=1}^{k} \varepsilon_i^2 + \frac{\varepsilon_g^2}{28.04 \cdot \log(1/\delta_g)} \right) \left( 1 + \frac{1}{2} \log \left( \frac{28.04 \cdot \log(1/\delta_g) \sum_{i=1}^{k} \varepsilon_i^2}{\varepsilon_g^2} + 1 \right) \right) \log(2/\delta_g)}.$$

We can then obtain a bound on the privacy loss with probability at least $1 - \delta_g/2$ over $\widetilde{v} \sim \widetilde{V}^0$

$$\left| \widetilde{\mathrm{Loss}}(\widetilde{v}) \right| \le \sum_{i=1}^{k} \widetilde{\mu}_i + \sqrt{2 \left( \sum_{i=1}^{k} \varepsilon_i^2 + \frac{\varepsilon_g^2}{28.04 \cdot \log(1/\delta_g)} \right) \left( 1 + \frac{1}{2} \log \left( \frac{28.04 \cdot \log(1/\delta_g) \sum_{i=1}^{k} \varepsilon_i^2}{\varepsilon_g^2} + 1 \right) \right) \log(2/\delta_g)}$$

$$\le \sum_{i=1}^{k} \varepsilon_i (e^{\varepsilon_i} - 1) + \sqrt{2 \left( \sum_{i=1}^{k} \varepsilon_i^2 + \frac{\varepsilon_g^2}{28.04 \cdot \log(1/\delta_g)} \right) \left( 1 + \frac{1}{2} \log \left( \frac{28.04 \cdot \log(1/\delta_g) \sum_{i=1}^{k} \varepsilon_i^2}{\varepsilon_g^2} + 1 \right) \right) \log(2/\delta_g)}.$$

$\square$

# 6   Advanced Composition for Privacy Odometers

One might hope to achieve the same sort of bound on the privacy loss from Theorem 2.10 when the privacy parameters may be chosen adversarially. However we show that this cannot be the case for any valid privacy odometer. In particular, even if an adversary selects the same privacy parameter $\varepsilon = o(\sqrt{\log(\log(n)/\delta_g)/k})$ each round but can adaptively select a time to stop interacting with `AdaptParamComp` (which is a restricted special case of the power of the general adversary – stopping is equivalent to setting all future $\varepsilon_i, \delta_i = 0$), then we show that there can be no valid privacy odometer achieving a bound of $o(\varepsilon\sqrt{k\log(\log(n)/\delta_g)})$. This gives a separation between the achievable bounds for a valid privacy odometers and filters. But for privacy applications, it is worth noting that $\delta_g$ is typically set to be (much) smaller than $1/n$, in which case this gap disappears (since $\log(\log(n)/\delta_g) = (1 + o(1))\log(1/\delta_g)$ ).

**Theorem 6.1.** *For any $\delta_g \in (0, O(1))$ there is no valid* $\text{COMP}_{\delta_g}$ *privacy odometer where*

$$\text{COMP}_{\delta_g}(\varepsilon_1, 0, \cdots, \varepsilon_k, 0) = \sum_{i=1}^{k} \varepsilon_i \left( \frac{e^{\varepsilon_i} - 1}{e^{\varepsilon_i} + 1} \right) + o\left( \sqrt{\sum_{i=1}^{k} \varepsilon_i^2 \log(\log(n)/\delta_g)} \right) \tag{13}$$

In order to prove Theorem 6.1, we use the following anti-concentration bound for a sum of random variables.

**Lemma 6.2** (Lemma 8.1 in [LT91]). *Let $X_1, \cdots, X_k$ be a sequence of mean zero i.i.d. random variables such that $|X_1| < a$ and $\sigma^2 = \mathbb{E}\left[X_1^2\right]$. For every $\alpha > 0$ there exists two positive constants $C_\alpha$ and $c_\alpha$ such that for every $x$ satisfying $\sqrt{k}\sigma C_\alpha \leq x \leq c_\alpha \frac{k\sigma^2}{a}$ we have*

$$\mathbb{P}\left[ \sum_{i=1}^{k} X_i \geq x \right] \geq \exp\left[ -(1+\alpha)\frac{x^2}{2k\sigma^2} \right]$$

For $\gamma \in [1/2, 1)$, we define the random variables $\xi_i \in \{-1, 1\}$ where

$$\mathbb{P}[\xi_i = 1] = \gamma \quad \mathbb{P}[\xi_i = -1] = 1 - \gamma. \tag{14}$$

Note that $\mathbb{E}[\xi_i] \overset{\text{def}}{=} \mu = 2\gamma - 1$ and $\mathbb{V}[\xi_i] \overset{\text{def}}{=} \sigma^2 = 1 - \mu^2$. We then consider the sequence of i.i.d. random variables $X_1, \cdots, X_n$ where $X_i = (\xi_i - \mathbb{E}[\xi_i])$. We denote the sum of $X_i$ as

$$M_n = \sum_{i=1}^{n} X_i. \tag{15}$$

We then apply Theorem 6.2 to prove an anti-concentration bound for the martingale given above.

**Lemma 6.3** (Anti-Concentration). *Consider the partial sums $M_t$ defined in (15) for $t \in [n]$. There exists a constant $C$ such that for all $\delta \in (0, O(1))$ and $n > \Omega\left( \log(1/\delta) \cdot \left( \frac{1+\mu}{\sigma} \right)^2 \right)$ we have*

$$\mathbb{P}\left[ \exists t \in [n] \text{ s.t. } M_t \geq C\sigma\sqrt{t\log(\log(n)/\delta)} \right] \geq \delta.$$

*Proof.* By Theorem 6.2, we know that there exists constants $C_1, C_2, C_3$ and large $N$ such that for all $m > N \cdot \left(\frac{1+\mu}{\sigma}\right)^2$ and $x \in \left[1, C_3 \sqrt{m} \frac{\sigma}{1+\mu}\right]$, we have

$$\mathbb{P}\left[\sum_{i=1}^{m} X_i \geq C_1 \sqrt{m} \sigma x\right] \geq e^{-C_2 x^2}.$$

Rather than consider every possible $t \in [n]$, we consider $j \in \left\{m_\delta, m_\delta^2, \cdots, m_\delta^{\lfloor \log_{m_\delta}(n) \rfloor}\right\}$ where $m_\delta \in \mathbb{N}$ and $m_\delta > m \log(1/\delta)$. We then have for a constant $C$ that

$$\mathbb{P}\left[\exists t \in [n] \text{ s.t. } M_t \geq C\sigma \sqrt{t \log(1/\delta)}\right] \geq \mathbb{P}\left[\exists j \in \left[\lfloor \log_{m_\delta}(n) \rfloor\right] \text{ s.t. } M_{m_\delta^j} \geq C\sigma \sqrt{m_\delta^j \log(1/\delta)}\right]$$

$$= \sum_{j=1}^{\lfloor \log_{m_\delta}(n) \rfloor} \mathbb{P}\left[M_{m_\delta^j} \geq C\sigma \sqrt{m_\delta^j \log(1/\delta)} \,\middle|\, M_{m_\delta^\ell} \leq C\sigma \sqrt{m_\delta^\ell \log(1/\delta)} \quad \forall \ell < j\right]$$

$$\geq \sum_{j=1}^{\lfloor \log_{m_\delta}(n) \rfloor} \mathbb{P}\left[M_{m_\delta^j} \geq C\sigma \left(\sqrt{m_\delta^j \log(1/\delta)} + \sqrt{m_\delta^{j-1} \log(1/\delta)}\right)\right]$$

$$= \sum_{j=1}^{\lfloor \log_{m_\delta}(n) \rfloor} \mathbb{P}\left[M_{m_\delta^j} \geq C\sigma \left(1 + 1/\sqrt{m_\delta}\right) \sqrt{m_\delta^j \log(1/\delta)}\right]$$

$$\geq \sum_{j=1}^{\lfloor \log_{m_\delta}(n) \rfloor} \mathbb{P}\left[M_{m_\delta^j} \geq 2C\sigma \sqrt{m_\delta^j \log(1/\delta)}\right]$$

Thus, we set $C = \frac{C_1}{2\sqrt{C_2}}$ and then for any $\delta$ such that $\sqrt{C_2} < \sqrt{\log(1/\delta)} < C_3 \sqrt{C_2} \sqrt{m_\delta} \frac{\sigma}{1+\mu}$, we have

$$\mathbb{P}\left[\exists t \in [n] \text{ s.t. } M_t \geq C\sigma \sqrt{t \log(1/\delta)}\right] \geq \lfloor \log_{m_\delta}(n) \rfloor \delta.$$

$\square$

---

**Algorithm 5** Stopping Time Adversary $\mathcal{A}_{\varepsilon,\delta}$ with constant $C$

---

    **for** $i = 1, \cdots, k$ **do**
        $\mathcal{A}_{\varepsilon,\delta} = \mathcal{A}_{\varepsilon,\delta(C, Y_1, \cdots, Y_{i-1})}$ gives datasets $\{0, 1\}$, parameter $(\varepsilon, 0)$ and $\mathtt{RR}_\varepsilon$ to $\mathtt{AdaptParamComp}$.
        $\mathcal{A}_{\varepsilon,\delta}$ receives $Y_i \in \{\top, \bot\}$.
        **if** $Y_i = \top$ **then**
            $X_i = \varepsilon$
        **else**
            $X_i = -\varepsilon$
        **if** $\sum_{j=1}^{i}\left(X_j - \varepsilon \frac{e^\varepsilon - 1}{e^\varepsilon + 1}\right) \geq C\left(\varepsilon \sqrt{t \log(\log(n)/\delta)}\right)$, **then**
            $\varepsilon_{i+1}, \cdots \varepsilon_k = 0$
        BREAK

---

We next use Theorem 6.3 to prove that we cannot have a bound like Theorem 2.10 in the adaptive privacy parameter setting, which uses the *stopping time adversary* given in Algorithm 5.

*Proof of Theorem 6.1.* Consider the stopping time adversary $\mathcal{A}_{\varepsilon,\delta_g}$ from Algorithm 5 for a constant $C$ that we will determine in the proof. Let the number of rounds $k = n$ and $\varepsilon = 1/n$. In order to use Theorem 6.3 we define $\gamma = \frac{e^\varepsilon}{1+e^\varepsilon}$ from (14). Because we let $\varepsilon$ depend on $n$, we have $\mu \equiv \mu_n = \frac{e^{1/n}-1}{e^{1/n}+1} = O(1/n)$ and $\sigma \equiv \sigma_n = 1 - \mu_n^2 = 1 - O(1/n^2)$ which gives $\frac{1+\mu_n}{\sigma_n} = \Theta(1)$. We then relate the martingale in (15) with the privacy loss for this particular adversary in AdaptParamComp$(\mathcal{A}_{\varepsilon,\delta_g}, n, 0)$ with view $V$ who sets $X_t = \pm\varepsilon$ each round,

$$\sum_{j=1}^{t}\left(X_j - \frac{\mu_n}{n}\right) = \frac{1}{n}M_t \qquad \forall t \in [n].$$

Hence, at any round $t$ if $\mathcal{A}_{\varepsilon,\delta_g}$ finds that

$$\frac{1}{n}M_t \geq C\left(\frac{1}{n}\sqrt{t\log(\log(n)/\delta_g)}\right) \tag{16}$$

then she will set all future $\varepsilon_i = 0$ for $i > t$. To find the probability that (16) holds in any round $t \in [n]$ we use Theorem 6.3 with the constant $C$ from the lemma statement to say that (16) occurs with probability at least $\delta_g$.

Assume that $\texttt{COMP}_{\varepsilon_g}$ is a valid privacy odometer and (13) holds. We then know that with probability at least $1 - \delta_g$ over $v \sim V^b$ where $V^b$ is the view for AdaptParamComp$(\mathcal{A}_{1/n,\delta_g}, n, b)$

$$|\texttt{Loss}(v)| \leq \texttt{COMP}_{\delta_g}(\varepsilon_1, 0, \cdots, \varepsilon_k, 0)$$

$$\implies \left|\sum_{i=1}^{t}\texttt{Loss}_i(v_{\leq i})\right| = t \cdot \frac{\mu_n}{n} + o\left(\frac{1}{n}\sqrt{t\log\left(\frac{\log(n)}{\delta_g}\right)}\right) \qquad \forall t \in [n]$$

But this is a contradiction given that the bound in (16) at any round $t \in [n]$ occurs with probability at least $\delta_g$. $\qquad\square$

We now utilize the bound from Theorem 4.3 to obtain a concentration bound on the privacy loss.

**Lemma 6.4.** $\texttt{COMP}_{\delta_g}$ *is a valid privacy odometer for* $\delta_g \in (0, 1/e)$ *where* $\texttt{COMP}_{\delta_g}(\varepsilon_1, \delta_1, \cdots, \varepsilon_k, \delta_k) = \infty$ *if* $\sum_{i=1}^{k}\delta_i > \delta_g/2$ *and otherwise for any* $\beta > 0$,

$$\texttt{COMP}_{\delta_g}(\varepsilon_1, \delta_1, \cdots, \varepsilon_k, \delta_k) = \sum_{j=1}^{k}\varepsilon_j(e^{\varepsilon_j} - 1)/2 + \sqrt{2\left(\sum_{i=1}^{k}\varepsilon_i^2 + \beta\right)\left(1 + \frac{1}{2}\log\left(\frac{\sum_{i=1}^{k}\varepsilon_i^2}{\beta} + 1\right)\right)\log(2/\delta_g)}.$$

*Proof.* We will follow a similar argument as in Theorem 5.1 where we use the same martingale $\widetilde{M}_k$ from (11). We can then directly apply Theorem 4.3 to get the following for any $\beta > 0$ with probability at least $1 - \delta_g/2$

$$\left|\widetilde{M}_k\right| \leq \sqrt{2\left(\sum_{i=1}^{k}\varepsilon_i^2 + \beta\right)\left(1 + \frac{1}{2}\log\left(\frac{\sum_{i=1}^{k}\varepsilon_i^2}{\beta} + 1\right)\right)\log(2/\delta_g)}$$

$\qquad\square$

We next need to determine what value of $\beta > 0$ to choose in the above result. If we were to set $\beta = \sum_{i=1}^{k} \varepsilon_i^2$, then we would get asymptotically close to the same bound as in Theorem 2.10, however, the $\varepsilon_i$ are random variables, and their realizations cannot be used in setting $\beta$; further, we know from Theorem 6.1 that such a bound cannot hold in this setting.

We now give our main positive result for privacy odometers, which is similar to our privacy filter in Theorem 5.1 except that $\delta_g$ is replaced by $\delta_g/\log(n)$, as is necessary from Theorem 6.1. Note that the bound incurs an additive $1/n^2$ loss to the $\sum_i \varepsilon_i^2$ term that is present without privacy. In any reasonable setting of parameters, this translates to at most a constant-factor multiplicative loss, because there is no utility running any differentially private algorithm with $\varepsilon_i < \frac{1}{10n}$ (we know that if $A$ is $(\varepsilon_i, 0)$-DP then $A(\mathbf{x})$ and $A(\mathbf{x}')$ for any pair of inputs have statistical distance at most $e^{\varepsilon_i n} - 1 < 0.1$, and hence the output is essentially independent of the input - note that a similar statement holds for $(\varepsilon_i, \delta_i)$-DP.)

**Theorem 6.5** (Advanced Privacy Odometer). $\text{COMP}_{\delta_g}$ *is a valid privacy odometer for* $\delta_g \in (0, 1/e)$ *where* $\text{COMP}_{\delta_g}(\varepsilon_1, \delta_1, \cdots, \varepsilon_k, \delta_k) = \infty$ *if* $\sum_{i=1}^{k} \delta_i > \delta_g/2$, *otherwise if* $\sum_{i=1}^{k} \varepsilon_i^2 \in [1/n^2, 1]$ *then*

$$\text{COMP}_{\delta_g}(\varepsilon_1, \delta_1, \cdots, \varepsilon_k, \delta_k) = \sum_{i=1}^{k} \varepsilon_i \left( \frac{e^{\varepsilon_i} - 1}{2} \right) + 2 \sqrt{\sum_{i=1}^{k} \varepsilon_i^2 \left( 1 + \log\left(\sqrt{3}\right) \right) \log(4 \log_2(n)/\delta_g)}. \tag{17}$$

*and if* $\sum_{i=1}^{k} \varepsilon_i^2 \notin [1/n^2, 1]$ *then* $\text{COMP}_{\delta_g}(\varepsilon_1, \delta_1, \cdots, \varepsilon_k, \delta_k)$ *is equal to*

$$\sum_{i=1}^{k} \varepsilon_i \left( \frac{e^{\varepsilon_i} - 1}{2} \right) + \sqrt{2 \left( 1/n^2 + \sum_{i=1}^{k} \varepsilon_i^2 \right) \left( 1 + \frac{1}{2} \log\left( 1 + n^2 \sum_{i=1}^{k} \varepsilon_i^2 \right) \right) \log(4 \log_2(n))/\delta_g}. \tag{18}$$

*Proof.* We again focus on a valid privacy odometer for $\{\delta_i\} \equiv 0$ and the martingale $\widetilde{M}_k$ from Equation (11). We then discretize the choices of $\beta \in \mathcal{D}_n \overset{\text{def}}{=} \{1/n^2, 2/n^2, 4/n^2, \cdots, 1/2, 1\}$ in Theorem 6.4, and then take a union bound over all $\beta \in \mathcal{D}_n$ to say that for the martingale $\widetilde{M}_k$ in (11) the following holds with probability at least $1 - \delta_g$ simultaneously over all $\beta \in \mathcal{D}_n$

$$|\widetilde{M}_k| \le \sqrt{2 \left( \sum_{i=1}^{k} \varepsilon_i^2 + \beta \right) \left( 1 + \frac{1}{2} \log\left( \frac{\sum_{i=1}^{k} \varepsilon_i^2}{\beta} + 1 \right) \right) \log(2 \log_2(n^2)/\delta_g)}.$$

Thus, for each realization $\sum_{i=1}^{k} \varepsilon_i^2 \in [1/n^2, 1]$, we can select $\beta$ to be the largest value in $\mathcal{D}_n$ that is just below $\sum_{i=1}^{k} \varepsilon_i^2$, i.e. $1 \le \frac{\sum_{i=1}^{k} \varepsilon_i^2}{\beta} \le 2$. This then gives the following bound with probability at least $1 - \delta_g/2$ when $\sum_{i=1}^{k} \varepsilon_i^2 \in [1/n^2, 1]$,

$$|\widetilde{M}_k| \le \sum_{j=1}^{k} \sqrt{2 \left( 2 \sum_{i=1}^{k} \varepsilon_i^2 \right) \left( 1 + \frac{1}{2} \log(2 + 1) \right) \log(4 \log_2(n)/\delta_g)}.$$

For the bound given in (18), we set $\beta = 1/n^2$. Hence, we would have with probability at least $1 - \delta_g/2$ when $\sum_{i=1}^{k} \varepsilon_i^2 \notin [1/n^2, 1]$,

$$|\widetilde{M}_k| \le \sum_{j=1}^{k} \sqrt{2 \left( 1/n^2 + \sum_{i=1}^{k} \varepsilon_i^2 \right) \left( 1 + \frac{1}{2} \log\left( 1 + n^2 \sum_{i=1}^{k} \varepsilon_i^2 \right) \right) \log(4 \log_2(n)/\delta_g)}.$$

$\square$

In the above theorem, we only allow privacy parameters such that $\sum_{i=1}^{k} \varepsilon_i^2 \in [1/n^2, 1]$. This assumption is not too restrictive, since the output of a single ($\ll 1/n$)-differentially private algorithm is nearly independent of its input. More generally, we can replace $1/n^2$ with an arbitrary "granularity parameter" $\gamma$ and require that $\sum_{i=1}^{k} \varepsilon_i^2 \in [\gamma, 1]$. When doing so, $\log_2(n^2)/\delta_g$ in (17) will be replaced with $\log_2(1/\gamma)/\delta_g$. For example, we could require that $\varepsilon_1 \geq \delta_g$, in which case we can choose $\gamma = \delta_g^2$, which would not affect our bound substantially.

## Acknowledgements

The authors are grateful Jack Murtagh for his collaboration in the early stages of this work, and for sharing his preliminary results with us. We thank Andreas Haeberlen, Benjamin Pierce, and Daniel Winograd-Cort for helpful discussions about composition. We further thank Daniel Winograd-Cort for catching an incorrectly set constant in an earlier version of Theorem 5.1.

## Footnotes

[1] A *statistical query* is parameterized by a predicate $\phi$, and asks "how many elements of the dataset satisfy $\phi$?" Changing a single element of the dataset can change the answer to the statistical query by at most 1.

[2]The same analysis holds for hetereogeneous parameters $(\varepsilon_1, \ldots, \varepsilon_k)$ are used in each round as long as they are all fixed in advance. For basic composition $\varepsilon k$ is replaced with $\sum_{i=1}^{k} \varepsilon_i$ and for advanced composition $\varepsilon\sqrt{k}$ is replaced with $\sqrt{\sum_{i=1}^{k} \varepsilon_i^2}$.

[3]Note that in the adaptive parameter composition game, the adversary has the option of effectively stopping the composition early at some round $k' < k$ by simply setting $\varepsilon_i = \delta_i = 0$ for all rounds $i > k'$. Hence, the parameter $k$ will not appear in our composition theorems the way it does when privacy parameters are fixed. This means that we can effectively take $k$ to be infinite. For technical reasons, it is simpler to have a finite parameter $k$, but the reader should imagine it as being an enormous number(say the number of atoms in the universe) so as not to put any constraint at all on the number of rounds of interaction with the adversary.