[Reviews · NeurIPS 2016]

Reviewer 1

Summary

In the adaptive composition of differentially private mechanisms, this study considers a setting that, the privacy budget \epsilon and \delta can be adaptively changed after observing the responses of mechanisms at each round. To achieve differential privacy in this pay-as-you-go setting, two notions are introduced, privacy filter and privacy odometer. The privacy filter is a function that determines whether or not to stop releasing responses to analysts at each round. The privacy odometer is a function that evaluates the privacy budget consumed at the timing of evaluation without knowledge of the entire privacy budget that would be consumed in the future. In the pay-as-you-go-setting, the adaptive composition theorem is not exactly same as the regular adaptive setting, but incurs at most a constant-factor multiplicative loss.

Qualitative Assessment

The problem discussed in this paper is novel and very interesting theoretically. It was unclear to me, in what kind of situation, the adaptive parameter selection is necessary. In regular differential privacy settings, analysts issue statistical queries and the curator controls the entire privacy budget so that the entire privacy breach does not exceed a given privacy budget. In the pay-as-you-go setting, who determines the privacy budget, the curator or analysts? If the analysts can determine the privacy budget at each round, what if the analyst does not listen to the advice of the privacy filter? If the curator can determine the entire privacy budget, why does she need to run it in the pay-as-you-go setting? It seems to be difficult to understand the structure of the paper without supplementary. For example, it is unclear why Section 4 is necessary. The readers of this paper would be familiar with the notion of differential privacy and adaptive composition. To me, it was hard to me to understand the intention of Lemma 3.3. My recommendation is to spend more space on the main contribution of this work than preliminaries on differential privacy.

Confidence in this Review

1-Less confident (might not have understood significant parts)


Reviewer 2

Summary

The composition theorem in differential privacy estimates the privacy cost resulting from running several differentially private computations, possibly adaptively. As the most common way of designing differentially private mechanisms for complex tasks is to put together basic DP building blocks, the composition theorem is the all-important glue to hold it all together. The traditional composition theorem works when we run a sequence of DP mechanisms where the privacy cost of each step, and the number of steps are fixed in advance. In some settings, it is important to be able to run adaptively: end early in some cases and go home with a smaller privacy loss, or use different privacy parameters per iteration depending on the answers so far. This is the question the authors study in this work. The authors show that the failure of previous composition theorems to apply to this setting is not just a technical omission. The adaptivity in fact comes at an asymptotic cost: if the adversary runs for a DP mechanism for a number k of steps that she chooses, the privacy loss can be made at least as large as sqrt{k loglog k}, whereas for fixed k, this loss is sqrt{k}. The paper shows that one can still recover good bounds: the sqrt loglog k loss is the worst possible: the authors show an upper bound. Moreover, if the goal is to only allow running until some privacy budget is exhausted (but not publish a privacy cost at each step), then one only loses a constant over the standard strong composition theorem. Finally, the authors also show that the standard composition theorem does not lose anything. This is valuable as for small k, it gives better bounds than the strong composition theorem.

Qualitative Assessment

The paper relates to an interesting and very important question in differential privacy. It extends the applicability of composition theorems beyond what was known earlier, and does so with essentially optimal loss in parameters. Given the importance of privacy to the NIPS community, I think this is a solid contribution that should be accepted.

Confidence in this Review

3-Expert (read the paper in detail, know the area, quite certain of my opinion)


Reviewer 3

Summary

This paper mainly considers adaptive composition theorem for differential privacy. The paper allows privacy parameters and number of rounds to be adaptive, and to solve the corresponding problem (for example, differential privacy is not defined suitably), authors propose two natural concepts, the privacy odometer and privacy filter. The privacy odometer can give an upper bound of privacy loss on every round, and the privacy filter can determine whether to continue or to halt the process, in the guarantee of global privacy budget. What’s more, authors prove the privacy filter has the same asymptotic order of bound as advanced composition theorem (C. Dwork 2010), while this cannot be achieved by the privacy odometer, except with an addition O(\sqrt{\log \log(n)}) factor.

Qualitative Assessment

In my personal view, the situation considered in this paper is much more general than almost all the existing work, and is more appropriate to use if differential privacy is applied in real word. Besides it, authors also give a deep understanding for the proposed primitives. These are main reasons for me to accept or to rate high scores for this paper, and I think it may open another door for differential privacy related research. Intuitively speaking, giving an upper bound for privacy loss every round is more difficult than determining whether to continue or to halt the process, as we can easily obtain a privacy filter from a privacy odometer. Thus, I think conclusions in this paper are theoretical sound. A question I am concerned is whether we can obtain a privacy odometer from a privacy filter, for example, once we can solve equation (7) to put \epsilon_g in only one side, could we say the corresponding term in the other side can be used as a privacy odometer? Some minor comments: 1. For the last word in line 86, I think authors want to say “Continue” not “Halt”; 2. In definition 2.6, there is a clerical error with the third case in randomized response; 3. In line 278 – 279, it will be better to explain more for the content in the brackets, as I am not clear why the left side of the inequality is an upper bound of statistical distance, and why this can conclude \epsilon_i has to be larger than \frac{1}{10n} (though I know this is a basic fact for many DP algorithms); 4. The format of paper’s references and supplementary is different with NIPS standard format.

Confidence in this Review

2-Confident (read it all; understood it all reasonably well)


Reviewer 4

Summary

The paper investigates an interesting question about the adaptive composition of differentially private procedures when the differential privacy parameters themselves are chosen adaptively. That is, the paper studies the setting where the number of rounds of interaction with the differentially private mechanism is not set beforehand, and the privacy parameters in each round is decided adaptively based on the outcomes/settings of the previous interactions. Under such setting, the standard composition theorems are not necessarily valid. The paper provides a characterization of the composition under this setting using two objects: "privacy filter" and "privacy odometer" that the authors define. The main result shows a separation between composition in the parameters-adaptive and non-adaptive (standard) settings. This separation, however, is only by a small asymptotic factor.

Qualitative Assessment

The paper studies a new interesting setting in differential privacy. The authors give careful analysis that answers almost completely the composition question in this setting. However, despite being non-trivial, the results are not very surprising (though this has nothing to do with the quality of the work).

Confidence in this Review

3-Expert (read the paper in detail, know the area, quite certain of my opinion)


Reviewer 5

Summary

The paper proposes a new setting for composition in differential privacy, where the length of composition and the privacy parameters can be chosen adaptively. It proposes privacy odometer that returns the privacy loss up to certain time, and privacy filter as a stopping rule to detect if the privacy budget is exceeded. It then provides bounds for both, compares them with existing composition theorems, and shows separation between privacy odometer and privacy filter.

Qualitative Assessment

Composition for differential privacy is a very important topic in differential privacy. And it is very interesting to look into the setting when the privacy parameters are also chosen adaptively. The paper makes a significant contribution to the direction of this setting. The paper is well-written. Yet it is pretty dense and it will be better if the paper can be re-organized by moving more contents to the appendix.

Confidence in this Review

3-Expert (read the paper in detail, know the area, quite certain of my opinion)